# Oxidative Metabolism in Brain Ischemia and Preconditioning: Two Sides of the Same Coin

**DOI:** 10.3390/antiox13050547

**Published:** 2024-04-29

**Authors:** Elena D’Apolito, Maria Josè Sisalli, Michele Tufano, Lucio Annunziato, Antonella Scorziello

**Affiliations:** 1Division of Pharmacology, Department of Neuroscience Reproductive Sciences and Dentistry, Federico II University of Naples, 80131 Napoli, Italy; elena.dapolito@unina.it (E.D.); michele.tufano2@unina.it (M.T.); 2Department of Translational Medicine, Federico II University of Naples, 80131 Napoli, Italy; mariajose.sisalli@unina.it; 3IRCCS Synlab SDN S.p.A, Via Gianturco 113, 80143 Naples, Italy; lucio.annunziato@gmail.com

**Keywords:** mitochondria, cerebral ischemia, ischemic tolerance, ROS, RNS, neuroprotection, neurotoxicity

## Abstract

Brain ischemia is one of the major causes of chronic disability and death worldwide. It is related to insufficient blood supply to cerebral tissue, which induces irreversible or reversible intracellular effects depending on the time and intensity of the ischemic event. Indeed, neuronal function may be restored in some conditions, such as transient ischemic attack (TIA), which may be responsible for protecting against a subsequent lethal ischemic insult. It is well known that the brain requires high levels of oxygen and glucose to ensure cellular metabolism and energy production and that damage caused by oxygen impairment is tightly related to the brain’s low antioxidant capacity. Oxygen is a key player in mitochondrial oxidative phosphorylation (OXPHOS), during which reactive oxygen species (ROS) synthesis can occur as a physiological side-product of the process. Indeed, besides producing adenosine triphosphate (ATP) under normal physiological conditions, mitochondria are the primary source of ROS within the cell. This is because, in 0.2–2% of cases, the escape of electrons from complex I (NADPH-dehydrogenase) and III of the electron transport chain occurring in mitochondria during ATP synthesis leads to the production of the superoxide radical anion (O_2_^•−^), which exerts detrimental intracellular effects owing to its high molecular instability. Along with ROS, reactive nitrosative species (RNS) also contribute to the production of free radicals. When the accumulation of ROS and RNS occurs, it can cause membrane lipid peroxidation and DNA damage. Here, we describe the intracellular pathways activated in brain tissue after a lethal/sub lethal ischemic event like stroke or ischemic tolerance, respectively, highlighting the important role played by oxidative stress and mitochondrial dysfunction in the onset of the two different ischemic conditions.

## 1. Introduction

Oxidative stress is a state that occurs when there is an imbalance between the generation of ROS and the antioxidant mechanisms within the cell [1,2]. Its was first defined back in 1985 by Helmut Sies [3], and it has been a topic of interest ever since. The reason behind this interest lies in the involvement of oxidative stress in several neurological disorders, including multiple sclerosis, amyotrophic lateral sclerosis, Parkinson’s disease, Alzheimer’s Disease, and ischemic stroke [4,5]. Moreover, oxidative stress is associated with mitochondrial dysfunction since these organelles play a key role in regulating oxidative phosphorylation, ATP production, and, consequently, cellular metabolism. Data obtained in our laboratory demonstrated that mitochondrial dysfunction correlates with changes in mitochondrial morphology and mitochondrial quality control in hypoxic conditions [6,7]. Interestingly, in ischemic conditions, a reduction in oxygen supply promotes the functional impairment of mitochondria, a hallmark of ischemic injury progression, since, as the damage occurs in the brain, it evolves into ROS production [8]. Recently, the relationship between mitochondrial dynamics and oxidative stress has received great interest because, in addition to being the main source of ROS within the cell, it is also responsible for cellular health. Indeed, to ensure cellular balance, mitochondria undergo constant fission and fusion, and furthermore, through the activation of mitophagy, the maintenance of a functional mitochondrial network in the cell is preserved thanks to the removal of damaged mitochondria and the activation of mitochondrial biogenesis [7,9]. The coordination between these two opposite processes, mitophagy and mitochondrial biogenesis, regulates mitochondrial quantity and quality in response to cellular metabolic state and to other intracellular or environmental signals, such as oxidative stress. Over the years, several in vitro and in vivo models of stroke have been described in order to investigate the molecular mechanisms leading to mitochondrial dysfunction and oxidative stress and understand what the intracellular consequences related to free radical production promoted by metabolic stress conditions are [10,11,12]. On the other hand, several studies have been performed in order to demonstrate that oxidative stress may exert beneficial effects [13,14,15,16] in experimental conditions characterized by sublethal stress conditions [13,14,15,16]. In this context, ischemic preconditioning (IP), an experimental model that recalls the tolerance triggered by short ischemic episodes defined in humans as TIA, has received particular attention since it represents a condition in which the exposure to an ischemic insult of brief intensity improves tolerance towards a subsequent ischemic event [17]. This model, reproduced both in in vivo and in vitro [18,19,20,21], has been very helpful in unravelling the main actors of the ischemic cascade and studying the consequential neuroprotective mechanisms activated in the preconditioned cells (Table 1). Indeed, it could be used to identify possible drug targets for the future for the development of innovative therapeutic strategies for the treatment of cerebral ischemia, particularly with regard to mitochondrial dysfunction and free radical production.

Among the different intracellular pathways activated by IP, mitochondria have been identified as promising neuroprotective players due to their ability to modulate the production of free radicals like ROS and nitric oxide (NO) (Table 1), which, in small amounts, behave as positive regulators of cell survival [16].

In this review, in addition to exploring the detrimental role played by ROS and RNS produced during an ischemic insult, we will discuss their neuroprotective role following IP in order to describe two processes related to ROS/RNS-regulated intracellular pathways that differ depending on the intensity and type of conditions, drawing attention to the important relationship that free radicals and oxidative stress establish with mitochondrial homeostasis in both experimental conditions.

## 2. Oxidative Stress: What Is It, and How Does It Affect Cellular Health?

The pathological role of oxidative stress in the onset of neurodegenerative disorders is related to the elevated uptake of molecular oxygen (O_2_) in brain tissue, to its high concentrations of lipids and abundance of unsaturated fatty acids [36], and to its low antioxidant capacity [37]. Above all, oxygen is the key player in the mitochondrial OXPHOS pathway, in which ROS synthesis can occur as a physiological side-product of the process. Indeed, besides producing ATP under normal physiological conditions, mitochondria are the primary source of ROS within the cell [38,39,40]. This occurs because O_2_ is the final acceptor of electrons, however, in 0.2–2% of cases, the escape of electrons from complex I (NADPH-dehydrogenase) and III of the electron transport chain occurring in mitochondria during ATP synthesis leads to the production of the superoxide radical anion (O_2_^•−^) [41]. Besides O_2_^•−^, among the ROS molecules worth mentioning are the hydroxyl radical (OH–) and hydroperoxyl radical (HOO•). ROS molecules are characterized by the presence of at least one unpaired electron, which makes them extremely unstable. In addition to mitochondria, several other sources of ROS may be detected in different cellular compartments like plasma membranes, peroxisomes, the smooth endoplasmic reticulum, and the cytosol, where intracellular reactions involving different enzymes and transition metals [2,42,43] take place. Along with ROS, reactive nitrosative species (RNS) also contribute to the production of free radicals. Specifically, the major source of RNS in cells is represented by NO, which is produced by three different isoforms of the nitric oxide synthase enzymes (NOSs), namely the endothelial (eNOS), the neuronal (nNOS), and the inducible (iNOS) forms [22,37]. NO exerts either beneficial or detrimental effects within the cell depending on its concentration. Indeed, at low concentrations, it regulates the activity of protein kinases, phosphatases, transcription factors, and gene expression through the activation of guanylate cyclase and the production of the intracellular signaling messenger cyclic GMP (cGMP), thus playing a neuroprotective effect. Indeed, cGMP activates the protein kinase B (Akt) and cAMP response element-binding protein (CREB) pathway, inducing an antiapoptotic and pro-survival effect. Moreover, an increase in intracellular cGMP levels may modify neurotransmission, synaptic plasticity, and brain activity since NO is also involved in the S-nitrosylation of NR1 and NR2 subunits of the N-methyl-D-aspartate receptor (NMDA), a process that inactivates the receptor, thus reducing the intracellular calcium concentration [22]. On the other hand, the most toxic derivative of NO, peroxynitrite (ONOO–), derived from the combination of NO with O_2_^•−^, is unstable and can be converted into other reactive species, such as the nitrile radical (NO_2_^−^), which is responsible for the hydroxylation and nitration of proteins residues [43,44]. Moreover, ONOO- and O_2_^•−^ have detrimental effects on mitochondria. Indeed, they cause mitochondrial fragmentation [45,46], leading to the impaired detoxification of ROS, the loss of mitochondrial membrane potential, impaired oxidative metabolism, the release of pro-apoptotic factors, and further ROS generation, thus creating a vicious cycle and massive apoptosis activation [47]. ONOO-, apart from activating other enzymes generating ROS, such as xanthine oxidase, is also responsible for the nitration of Try34 on mitochondrial MnSOD (SOD2), thus inhibiting its function [48,49]. Superoxide dismutase (SOD) enzymes are the major ROS-detoxifying enzymes inside the cell. There are three different isoforms: cytoplasmic Cu/ZnSOD (SOD1), mitochondrial MnSOD (SOD2), and the extracellular Cu/ZnSOD (SOD3) [49]. MnSOD is located within the mitochondrial matrix that catalyzes the conversion of O_2_^•−^ into hydrogen peroxide (H_2_O_2_) and O_2_ [50,51,52], which is subsequently converted into H_2_O as a result of catalase, peroxiredoxin, or glutathione peroxidase action [53,54]. Therefore, MnSOD inhibition will lead to peroxynitrite accumulation and a consequential increase in oxidative stress, triggering apoptosis [55]. Apart from MnSOD, other intracellular pathways might be involved in preserving redox equilibrium. Among them, it is worth mentioning the nuclear factor erythroid 2-related factor 2 (Nrf2) [37] due to its ability, once activated, to exert neuroprotection in stress conditions and in ischemic stroke [56,57,58]. Nrf2 is a transcriptional factor able to bind the antioxidant response element (ARE) in the nucleus and consequently induce antiapoptotic pathways [59] to reduce calcium overload [60] and exert anti-inflammatory effects [61], thus reducing oxidative stress. Nrf2 activity its regulated by a cytosolic protein, Keap1 (Kelch-like ECH-associated protein 1), which is sensitive to ROS intracellular concentrations. Indeed, when ROS levels are low, Nrf2 and Keap1 interaction causes Nrf2 to be blocked at the cytosol level, whereas, when ROS production increases, Nrf2 translocates into the nucleus, leading to the expression of antioxidant genes such as SODs and Heme oxygenase-1 (HO-1) [62], which is also important for anti-inflammatory and antiapoptotic intracellular effects. In addition, it has also been demonstrated that Nrf2 plays an important role in the regulation of mitochondrial dynamics. Indeed, Nrf2 is able to increase the expression levels of nuclear respiratory factor-1 (NRF-1) [63] and mitochondrial transcription factor A (TFAM), which is responsible for mtDNA transcription [63,64]. For this reason, when there is an increase in ROS production for a reason other than direct mitochondrial damage, an indirect mechanism induced by Nrf2 targets inhibition, leading to mitochondrial damage. Considering the fundamental role played by mitochondria in cells, it is important to briefly highlight the mechanisms involved in mitochondrial homeostasis, which are impaired by oxidative stress. In this regard, the main leading actor of the fission process is a GTPase known as dynamin-related protein 1 (DRP1), a cytosolic protein recruited into the mitochondria by its adapters present on the outer mitochondrial membrane (OMM), such as mitochondrial fission protein 1 (Fis1) [65]. On the other hand, Mitofusin 1 and 2 (MFN1 and MFN2), GTPases located on the outer mitochondrial membrane (OMM), and Optic atrophy 1 (OPA1), a GTPase located on the inner mitochondrial membrane (IMM), are the main proteins involved [66,67]. Because increased ATP production is linked to mitochondrial fusion, while the inhibition of fusion is linked to impaired OXPHOS, mtDNA damage, and ROS production [68], the activity of the above-mentioned proteins is extremely important to guarantee the maintenance of a healthy mitochondrial network to ensure the correct response to oxidative stress within the cell.

Therefore, the range of ROS and RNS is directly correlated to cellular and antioxidant mechanism integrity. Indeed, depending on the severity of the disruption of the antioxidant defense, which is more pronounced in ischemia than in ischemic preconditioning, some intracellular pathways sensitive to ROS production can be activated to promote detrimental (necrosis/apoptotic cell death)/beneficial (cell survival) effects, respectively.

Considering the dual role of ROS and RNS (Table 2) in the regulation of oxidative metabolisms and cell survival [41,69], in the following paragraphs, we will clarify the contribution of ROS and RNS in pathological conditions closely related to oxidative stress and mitochondrial dysfunction, like brain ischemia and ischemic preconditioning.

## 3. The Role of Oxidative Stress and Mitochondria during Ischemic Brain Damage: The Detrimental Side of the Phenomenon

Oxidative stress and mitochondrial dysfunction are among the hallmarks of ischemic stroke, as they are responsible for the activation of different deleterious pathways that cause irreversible cellular damage. Stroke represents a public health problem worldwide, as it is the third leading cause of death and disability, following cardiovascular diseases and cancer [70]. Stroke can be classified into two types: hemorrhagic and ischemic. Ischemic stroke is the most frequent one, occurring when there is an obstruction of arterial blood flow to the brain, causing blockage and a reduction in blood flow. If the ischemia lasts long enough, it can lead to neuronal loss, resulting in an ischemic stroke [71,72]. The only approved treatment for ischemic stroke is recombinant thrombolytic tissue plasminogen activator (rt-PA, Genentech Activase) [73]. When administered, rt-PA is able to promote the reperfusion of the ischemic brain [74]. However, the detrimental effects on brain activity following rapid reperfusion have been widely recognized in numerous clinical and experimental studies; indeed, it can lead to brain edema or hemorrhagic transformation, causing significant neuronal death [75]. In addition to the role played following the ischemic event, oxidative stress also has a critical role in reperfusion injuries because the sudden increase in oxygen supply in the brain tissue can cause an oxidative burst [76,77]. Following ischemic damage, a chain of events is triggered by the detrimental ATP synthesis, such as an increase in [Na^+^] and [Ca^2+^] that occur inside the cell. This ionic imbalance is associated with an increase in the extracellular levels of glutamate, causing excitotoxicity that is also responsible for mitochondrial dysfunction and further calcium release from the mitochondria itself [23]. Excessive glutamate release causes the mass stimulation of NMDA receptors, whose activation may be involved in ROS production by inducing uncoupling neuronal mitochondrial electron transport [24,36]. Ca^2+^ dysregulation can also be caused by other channels, ion pumps, and exchangers such as the Na/Ca^2+^ exchanger (NCX) [25]. Among the three isoforms of NCX, only NCX3 was shown to be localized on mitochondria, specifically on the OMM, in [78]. When the dysfunction of two ATP-dependent plasma membrane pumps, Na^+^/K^+^ ATPase and Ca^2+^ ATPase, occurs in hypoxic conditions, NCX plays a crucial role in maintaining intracellular homeostasis by reinforcing the extrusion of Na^+^ ions and the influx of Ca^2+^. Although, in the initial phase of anoxia, the exchanger causes an increase in [Ca^2+^]_i_, its effect could be beneficial for neurons because it helps to decrease [Na^+^]i overload, which would otherwise lead to cellular swelling and sudden necrotic neuronal death. On the other side, in the later phase of neuronal anoxia, when [Ca^2+^]_i_ overload occurs, NCX may protect neurons from intracellular Ca^2+^ overload by lowering [Ca^2+^], thus protecting neurons from neurotoxicity and subsequent cell death [17]. The toxic role played by calcium in relation to oxidative stress has been demonstrated in an experimental model of oxygen–glucose deprivation (OGD) and reoxygenation. Indeed, the release of Ca^2+^ by activating nNOS is also responsible for NO synthesis, causing consequential neuronal damage in ischemic-like conditions [79]. NO overproduction will result in the S-nitrosylation of several substrates, including SOD2, MMP9 (Matrix Metalloproteinase 9), Parkin (decreasing its activity), and GAPDH. Furthermore, NO is able to activate cyclooxygenase (COX), causing consequential prostaglandin overproduction, leading to neuroinflammation [22,26]. At the mitochondrial level, following an ischemic event, the production of ROS leads to the opening of a Mitochondrial Permeability Transition Pore (mPTP) [27], which is associated with the further release of ROS and consequent activation of the apoptotic pathway by the release of cytochrome c (Cyt c) from mitochondria into the cytoplasm of neuronal cells [28,29,30]. This phenomenon, induced not only by oxidative stress but also by high levels of Ca^2+^ accumulation in the mitochondrial matrix, increases the permeability of the internal mitochondrial membrane, leading to the loss of mitochondrial membrane potential, with consequent alteration in OXPHOS and progressive ATP depletion [8]. Also, mitochondrial ROS contribute to the inflammatory response, activating immune cells through the MAPK and NF-kB pathways [31]. Three distinct phases of free radical generation were identified in hippocampal and cortical neurons exposed to OGD and reoxygenation. It has been shown that, in the early phase of OGD, there is an initial burst of mitochondrial ROS, the production of which appears to be subsequently replaced by the activation of xanthine oxidase. Finally, in the third phase, during the reoxygenation phase, the production of ROS, occurring once again, depends mainly on the action of nicotinamide adenine dinucleotide phosphate (NADPH) oxidase activated by [Ca^2+^] [80], since ROS molecules are also generated by NADPH oxidase, cyclooxygenases, and xanthine oxidase [81]. It has been shown that low levels of ROS and RNS do not cause mitochondrial damage, whereas high levels of ROS produced by ischemic damage affect mitochondrial morphology, causing mitochondria swelling or fragmentation. Experiments performed in fibroblasts [82] and in skeletal muscle myoblasts [83] exposed to exogenous H_2_O_2_ confirmed the role of oxidative stress in mitochondrial fragmentation and depolarization [45]. Moreover, it was also demonstrated that oxidative stress within the cell leads to the massive activation of Drp1, a marker of mitochondrial fission [84], thus explaining the excessive fragmentation observed in stress conditions. Interestingly, experiments performed in human endothelial cells (HUVECs) treated with growing amounts of hydrogen peroxide not only confirmed the tight relationship between oxidative stress and mitochondrial dysfunction but also demonstrated a dose-dependent detrimental effect of oxidative stress on mitochondria [85] in terms of increases in apoptosis, ROS production, and changes in mitochondrial morphology.

However, it is important to mention that mitochondria are highly dynamic organelles displaying an intrinsic ability to adapt to oxidative stress-like conditions and that this effect is time dependent. Indeed, as reported by Jendark and colleagues, mitochondrial morphology changed from fragmented after 24 h from H_2_O_2_ treatment to fused after 48 h of treatment, and the amount of fusion and fission events was almost identical in the treated and control cells [85]. The crucial role of ROS and RNS on mitochondrial dynamics is further empathized by the large number of studies showing that antioxidants specifically targeting mitochondria are able to reverse mitochondrial fragmentation induced by oxidative stress [86,87,88].

Therefore, it is possible to assume that mitochondria play the role of sensors that detect cellular oxidative stress since the accumulation of ROS/RNS influences mitochondrial dynamics to support the metabolic cellular conditions (Figure 1). In this scenario, it is reasonable to hypothesize that oxidative stress and ischemic damage can be ameliorated by reducing mitochondrial dysfunction through the regulation of mitochondrial quality control.

## 4. The Role of Oxidative Stress and Mitochondria during Cerebral Ischemic Preconditioning: The Neuroprotective Side of the Phenomenon

On the basis of the above considerations, the phenomenon of ischemic tolerance, also known as cerebral ischemic preconditioning (IPC), represents a useful model for investigating the molecular mechanisms involved in mitochondria’s adaptation to oxidative stress. Indeed, it has recently received particular attention due to its similarity to some pathological conditions, such as brief ischemic TIA, which appears to protect the brain against a subsequent lethal stroke [89,90]. It was described for the first time in the heart [91] and then in many other irrorated organs, including the brain [92]. It is a phenomenon that can be induced by a sublethal anoxic insult that improves tissue tolerance to a subsequent and possibly fatal ischemic event. Nowadays, IPC has been widely recognized as a relevant and effective experimental tool for understanding how the brain protects itself from ischemia, thus providing an innovative experimental model for the development of new neuroprotective strategies [17]. The peculiarity of ischemic tolerance is that it is triggered by the same pathways activated after an ischemic event, such as NMDA receptor activation, glutamate release, and ROS production [93]. Moreover, changes in gene expression have a main role, resulting in the activation of different intracellular pathways; indeed, after the preconditioning stimulus, unlike genes can be overexpressed or downregulated, leading to different outcomes [94]. The main pathways involved in neuroprotective mechanisms include mitogen-activated protein kinases (MAPKs), the PI3K-Akt pathway, and the protein kinase C-ε (PKCε) pathway [95,96].

PKCε has a role in mitochondrial OXPHOS and in the phosphorylation of mitochondrial respiratory-chain proteins. Furthermore, PKCε activity in mitochondria may also contribute to the regulation of mK^+^ ATP channels, which are important for preserving mitochondrial membrane potential, ensuring ATP production and reducing calcium influx [95,96]. Indeed, strong evidence supports the involvement of mitochondrial ROS and mK^+^ ATP channels in the neuroprotective mechanisms triggered by preconditioning [97,98]. To study the neuroprotective pathways, cerebral ischemic tolerance can be induced in vitro by a variety of sublethal insults, such as OGD, transient hypoxia, oxidative stress, hyperthermia, or heat shock. These are all stimuli able to protect neuronal cells in culture from a lethal ischemic insult if administered at least 22–24 h before the insult itself [18]. In vivo models of preconditioning can be reproduced through the induction of global ischemia, which affects the entire brain [19]; focal ischemia, usually caused by middle cerebral artery or single carotid artery occlusion, which is confined to a specific region of the brain; and forebrain ischemia, which is mainly induced by bilateral common carotid occlusion or four-vessel occlusion [20,21]. Among all, the transient occlusion of the middle cerebral artery (tMCAO) has been fundamental for demonstrating that a short period of tMCAO followed by a lethal ischemic episode is able to reduce the ischemic brain area [99]. Concerning the possibility of considering ischemic preconditioning as a model for investigating the dual roles of ROS and RNS in beneficial and detrimental effects on cell survival depending on the dosage and time of exposure, Ravati and colleagues observed that neuronal cultures pretreated with a moderate amount of ROS developed defenses against various types of neuronal damage and that the elimination of ROS stimulus during the preconditioning period reversed this form of protection when the cells were exposed to a subsequent lethal insult [13]. In line with these findings, it has been proven that NO, despite its neurotoxic role in ischemic conditions, is involved in the phenomenon of ischemic tolerance. Indeed, many studies from our laboratory and other laboratories have confirmed the emerging role of NO as one of the main activators of different pro-survival pathways in various experimental models, both in preconditioning and remote postconditioning [32,100,101]. In this regard, in preconditioned neurons, NO activates Ras and the downstream ERK1/2 pathway, which promotes cell survival and proliferation through the activation of MnSOD expression, as well as the Akt/PI3K pathway, thus promoting neuroprotection [14,15] (Figure 2). Interestingly, the exposure of preconditioned neurons to treatment with nNOS inhibitors caused a reduction in MnSOD expression and activity and abolished IPC-induced neuroprotection. Besides preconditioning, NO was also considered an important player in mediating neuroprotection in an experimental model of postconditioning, which raises more interest from a clinical point of view than preconditioning, considering its relevant translational potential. Postconditioning consists of the induction of a mild detrimental stimulus applied after a severe harmful event such as ischemic damage, which is able to induce protection in the tissue in which it is delivered. More interestingly, remote ischemic postconditioning reduces tissue injury caused by ischemia–reperfusion in distant organs [99]. Specifically, by inducing remote femoral postconditioning, Pignataro and colleagues demonstrated nNOS-induced neuroprotection, whereas eNOS or iNOS isoforms were not involved since selective inhibitors of these two isoforms of NOS did not prevent the neuroprotective effect induced by the remote postconditioning [32].

Another interesting aspect that links neuroprotection induced by ischemic tolerance to NO is its relationship with ionic homeostasis. As described before, intracellular Ca^2+^ accumulation, occurring in hypoxic conditions as a consequence of the hyperactivation of Ca^2+^ channels, represents the key trigger of excitotoxicity in neurons that cause mitochondrial damage, ROS release, and neuronal death. However, the increase in NO concentration observed in an in vivo model of brain ischemia induced hypoxia-inducible factor 1 (HIF-1) activation [102], which, in turn, was able to promote NCX1 gene upregulation, thus providing a pro-survival effect during ischemic preconditioning [33] (Figure 2). On the other hand, in vitro experiments performed on preconditioned cortical neurons exposed to OGD/reoxygenation/neuroprotection may occur through the modulation of calcium homeostasis in ER and mitochondria through NO/PI3K/Akt-mediated NCX1 and NCX3 upregulation [34]. These effects are in line with the regulatory properties of NCX and contribute to the belief that NO and ionic homeostasis are mechanisms responsible for IPC-induced neuroprotection. Indeed, NO and redox agents are able to affect the functional properties of the sodium calcium exchanger [103], thus preventing ionic homeostasis in stress conditions, which occurs in cerebral brain ischemia [104]. Through exploring these mechanisms, it becomes evident how important the tight relationship between the production of free radicals and intracellular calcium homeostasis is for guaranteeing neuroprotection in ischemic preconditioning. Indeed, the increase in NO that occurs in preconditioning is important as it allows for the influx of calcium within ER in order to prevent apoptosis related to ER-stress [34]. On the other hand, to preserve metabolic cellular conditions, it is extremely important to uphold mitochondrial calcium efflux, which allows for ATP production and neuronal survival [34]. Finally, the maintenance of mitochondrial function is fundamental for ensuring the proper activity of the mitochondrial quality control system, which is needed to make the cells less vulnerable to the oxidative insult [7].

In this regard, it is worth mentioning that NO produced in IPC might provide neuroprotection while also affecting mitochondrial dynamics (Figure 2). Concerning the molecular mechanisms involved in these effects, several hypotheses have been formulated [35]. Among them, one assumes that NO interacts with prohibitin (PHB), an important protein localized in the brain mitochondria that is involved in the regulation of mitochondrial morphology and dynamics [105]. By the way, Qu and colleagues, using the ischemia-reperfusion injury model, demonstrated that NO, interacting with PHB, induced S-nitrosylation on its Cys residue, which is essential for its ability to preserve neuronal viability under hypoxic stress conditions [106]. Similarly, the E3-ubiquitine ligase Siah2 plays a key role in preserving mitochondrial function and integrity in in vivo and in vitro models of ischemia, since, when activated in O2 impairment conditions, it translocates to depolarized mitochondria, where it induces the ubiquitination and the consequent proteasomal degradation of NCX3 and AKAP121, two mitochondrial proteins involved in the regulation of mitochondrial morphology and function [107]. In addition, Siah2, by orchestrating the balance between mitophagy and mitochondrial biogenesis in cortical neurons exposed to OGD/Reoxygenation, has a role in neuroprotection induced by ischemic preconditioning. This effect is tightly dependent on the redox status of the mitochondria [6,7]. All these neuroprotective pathways aimed at improving mitochondrial function are consistently correlated with better outcomes regarding the central role of mitochondria in regulating cellular health. For this reason, it is possible to speculate that the relationship between different amounts of ROS/RNS and the improvement of mitochondrial dynamics is crucial for explaining its possible role in different neuroprotective conditions.

## 5. Conclusions and Perspectives

The data discussed in this review have led us to conclude that brain ischemia and ischemic preconditioning represent two useful models for studying the consequences of oxidative stress and mitochondrial dysfunction on neuronal survival for different reasons.

Firstly, they allow us to demonstrate the tight relationship between free radical generation and mitochondrial dynamics. Indeed, depending on the intensity and the duration of O_2_ deprivation, different intracellular events happen in neurons, affecting their fate. Specifically, when the O_2_ supply is decreased for long periods, as is the case during ischemic events, the increase in the production of free radicals causes an abnormal increase in intracellular calcium (Table 1), which contributes to mitochondrial dysfunction and to their consequent fragmentation. These effects further stimulate ROS and RNS production, thus inducing cellular damage (Figure 1). Conversely, as described in Section 4, short and repeated periods of O_2_ deprivation, such as those occurring during ischemic preconditioning, are able to improve mitochondrial dynamics stimulated by low levels of NO production to allow for intracellular calcium efflux and promote neuroprotection (Figure 2). Therefore, it is possible to assert that both calcium and mitochondrial dysfunction represents two intracellular effectors of free radical production whose balance drives neuronal survival or death.

Secondly, these models emerged as useful tools that can be used to focus on the importance of the intrinsic properties of neuronal cells needed to face the detrimental intracellular effects of oxidative insult. Indeed, IPC has proven to be a useful model for the study of the endogenous mechanisms of neuroprotection in ischemic conditions.

Finally, due to their ability to reveal new molecular targets regulating the balance between neuroprotective and detrimental intracellular pathways, these models allow us to develop innovative therapeutically strategies based on the activation of mitochondrial quality control mechanisms or on the improvement of antioxidant intracellular defense in order to promote neuronal survival.

In conclusion, considering the dual role of ROS and RNS in the regulation of oxidative metabolisms and cell survival, it becomes extremely important to identify selective intracellular targets able to sense the oxidative imbalance and to restore the metabolic cellular status in pathological conditions like brain ischemia, which are highly dependent on oxidative stress and mitochondrial dysfunction.

In this regard, it is valuable to underline the potential therapeutic perspectives related to the modulation of oxidative stress in ischemic conditions, in which it is important to reduce mitochondrial free radical production by using common antioxidant drugs or by mimicking ischemic tolerance using drugs able to release free radicals like NO or limit ROS or RNS production in order to rescue the cells from oxidative injury. Therapeutic strategies [108,109,110,111,112,113] aimed at addressing these goals are reported in Table 3.

Moreover, it is worth underlining the potential therapeutically beneficial effects of melatonin [107] and tobacco’s role in neuroprotection [114]. Indeed, recent evidence suggests that the use of tobacco and its derivative compounds could prove to be a strategy able to mediate neuroprotective effects in neurodegenerative disorders such as Parkinson’s Disease and Alzheimer’s Disease [115,116]. This recent evidence allows us to hypothesize tobacco use as a potential preconditioning strategy in hypoxic/ischemic conditions. However, this finding needs to be investigated.

## Figures and Tables

**Figure 1 antioxidants-13-00547-f001:**
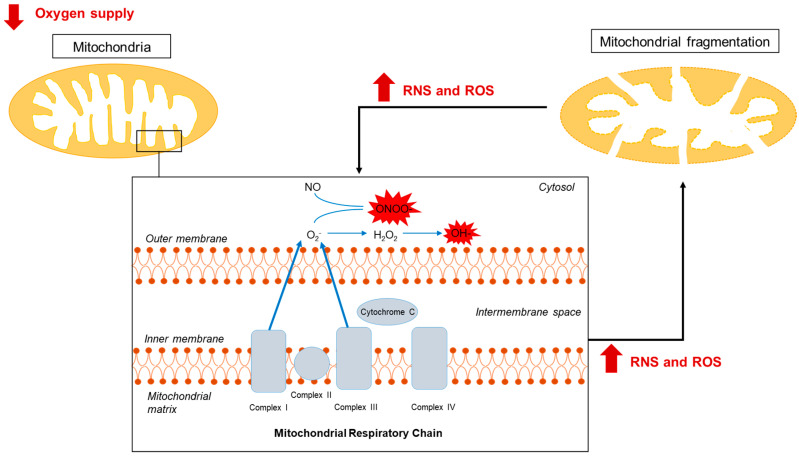
Hypoxia-induced mitochondrial dysfunction and free radical production. In ischemic/hypoxic conditions, the impairment of O_2_ supply causes the dysfunction of the mitochondrial respiratory chain, consequently increasing free radical production (ROS and RNS) and mitochondria fragmentation.

**Figure 2 antioxidants-13-00547-f002:**
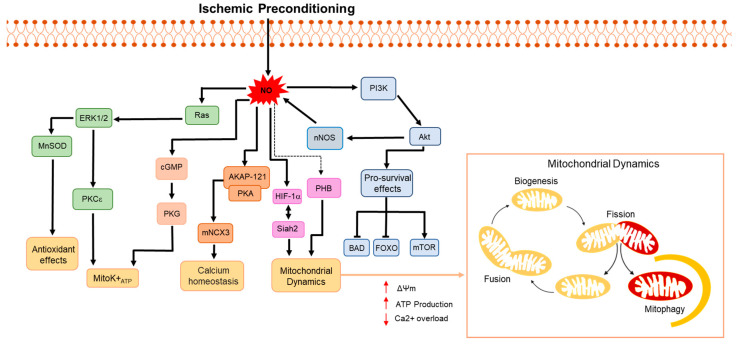
NO-induced intracellular pathways in ischemic preconditioning and their effects on mitochondrial dynamics. See text for the details.

**Table 1 antioxidants-13-00547-t001:** The key players involved in the ischemic cascade and in ischemic preconditioning.

Lethal/Sublethal Ischemic Event	Intracellular Outcomes	Intracellular Players
Ischemic Cascade	Detrimental [22,23,24,25,26,27,28,29,30,31]	Imbalance in Na^+^ and K^+^ ions Intracellular calcium overload Glutammate excitotoxicity Neuroinflammation mPTP opening Oxidative stress Apoptosis
Ischemic Preconditioning	Neuroprotective [7,14,15,16,32,33,34,35]	nNOS activation NO release NCX1 and NCX3 activation Mitochondrial calcium efflux HIF-1-induced genes E3-ubiquitin ligase Siah2 activation Mitophagy/Mitochondrial Biogenesis

**Table 2 antioxidants-13-00547-t002:** Intracellular events mediated by ROS and RNS depending on their cytosolic concentrations [22,43,44,45,46,47,48,49].

ROS and RNS Effects within the Cell	[ROS/RNS]_i_	Intracellular Pathways
Neurotrophic	Low concentrations	cGMP production Akt-CREB pathway S-Nytrosilation of NMDR subunits
Harmful	High concentrations	Mitochondrial damage Lipid peroxidation of membranes DNA damage Apoptosis

**Table 3 antioxidants-13-00547-t003:** Antioxidant therapy: recent updates [85,108,109,110,111,112,113].

Antioxidant Families	Drugs	Results
ROS/RNS-generating enzyme inhibitors	Apocynin Allopurinol Nitro-L-arginine methyl ester Nimesulide	Inhibits NOX family Inhibits Xanthine oxidase Inhibits NOS family Inhibits COX family
Flavonoids and polyphenolic	Resveratrol Ferulic acid Curcumin Polyphenol quercetin	Upregulate HO-1 and SOD Anti-inflammatory effects Antiapoptotic effects Inhibits ER stress
Hormones	Melatonin	Downregulates apoptosis Increases the expression of complexes I and IV of the ETC Blocks Fenton reaction Inhibits NF-κB activation Upregulates Nrf2 expression
Vitamins	α-Tocopherol (vitamin E)	Scavenger of lipid-soluble peroxyl radicals
	Ascorbic Acid (vitamin C)	Scavenger of water-soluble peroxyl radicals
Free radical scavenger	Edaravone	Inhibits peroxyl radicals’ oxidation of lipids
Mitochondria-targeted antioxidants	MitoQ SkQ1	Decreasing ROS production Increasing MnSOD activity Inhibits lipid peroxidation

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
