# Peer review of "Oxidative Metabolism in Brain Ischemia and Preconditioning: Two Sides of the Same Coin"

_antioxidants, 2024, doi:10.3390/antiox13050547_

Round 1
Reviewer 1 Report
The review manuscript highlights the dual nature of nitric oxide, which plays a physiological role in dilating arteries but can have detrimental effects in the presence of free radicals. Additionally, oxidative stress is discussed with its dual roles, exacerbating brain ischemia on one hand and serving as ischemic preconditioning to prevent exacerbation of brain ischemia on the other.
While the manuscript is well written, the content appears to be too broad and redundant, making it challenging to understand easily. Therefore, I suggest that the authors focus more specifically on the roles of nitric oxide and oxidative stress in the pathomechanisms of brain ischemia to streamline the manuscript and enhance clarity. It appears that the authors are not familiar with a difference between ischemia and hypoxia. Ischemia is not directly equal to hypoxia. Ischemia or hypoxia needs to be clearly discriminated when the authors intend to indicate something.
#1. As I commented it as a major comment, the manuscript has too much information beyond brain ischemia. Therefore, the authors have to delete some parts that are not relevant to the topic.
Ex)
On the line 99-101, ‘Moreover, different environmental conditions, such as radiation or pollution contribute to the increase in ROS production responsible for cellular detrimental effects [2, 24].’ However, it is unclear the connection between the above sentence and anteroposterior sentences. Consider delete them.
The contents on lines 133-164 are not applied to the following sections. I do not believe the contents are necessary to be described in the review. Consider delete them.
Are the contents on lines 231-252 associated with ischemic stroke? I think the contents just show the associations between oxidative stress and mitochondrial dysfunction independently of ischemic stroke.
#2. Several citations may not be appropriate.
Ref. 7 is not appropriate because it is not accepted in a scientific journal. Ref. 7 is not suitable to cite in the review. Furthermore, ‘Data obtained in our laboratory demonstrated that mitochondrial dysfunction correlates with changes in mitochondrial morphology and mitochondrial quality control in hypoxic conditions [6,7]. (Lines 43-45)’ and relevant other sentences should be completely deleted.
The authors cited Ref. 18 on the line 86. However, I could not find that neurodegenerative diseases are related to low antioxidant capacity in the affected brain to my best knowledge.
The authors cited ref. 84 on line 284. However, ref. 84 does not indicate the main pathways involved in neuroprotective mechanisms include mitogen-activated protein kinases (MAPKs), the PI3K-Akt pathway and 283 the protein kinase C-ε (PKCε) one. Cite an appropriate reference.
Ref. 104 on the line 350 does not show ‘among them one assumes that NO interacts with prohibitin (PHB), an important protein localized in brain mitochondria that is involved in the regulation of mitochondrial morphology and dynamics.’
#3. The authors may regard both hypoxia and ischemia as the same pathological conditions. Ischemic condition is different from hypoxic one. Therefore, the authors should focus on brain ischemia when ischemic pathologies are discussed, while the authors should focus on brain (ischemia-)hypoxia when (ischemic and) hypoxic pathologies are discussed.
#4. On lines 353-356, the authors mention that the E3-ubiquitin ligase Siah2 plays a key role in neuroprotection induced by ischemic preconditioning by orchestrating the balance between mitophagy and mitochondrial biogenesis in cortical neurons exposed to OGD/Reoxygenation. Does Siah2 plays a key role in neuroprotection via ubiquitination? Moreover, please appropriately cite after the above sentence. If Ref. 7 indicates the content of the above sentence, this is not the fact and the sentence have to be deleted.
#5. In Table 1, low concentration of ROS or RNS may lead to neuroprotective effects, but high ROS or RNS concentration may lead to deleterious effects. However, TIA is short-term temporal brain ischemia with relatively mild decrease of cerebral blood flow. Therefore, not only low or high concentration but also the duration may affect the pathomechanisms. The authors should include the angle of the time duration.
#6. Figure 2 appears to show that Siah2 and PHB may induce mitochondrial dynamics. However, contributions of Siah2 and PHB to mitochondrial fusion, fission, biogenesis, and mitophagy are not mentioned in the text.
Author Response
REFEREE #1
Comment # 1: “On the line 99-101, ‘Moreover, different environmental conditions, such as radiation or pollution contribute to the increase in ROS production responsible for cellular detrimental effects [2, 24].’ However, it is unclear the connection between the above sentence and anteroposterior sentences. Consider delete them.”
Answer: We agree with Referee’s suggestion to remove the sentence on line 99-101 since it is not properly related to the topic of review. See page 3, of the revised version of the manuscript
Comment # 2: The contents on lines 133-164 are not applied to the following sections. I do not believe the contents are necessary to be described in the review. Consider delete them.
Answer: We thank the Reviewer for the valuable observation on the contents described in Lines 133-164 of the manuscript that gave us the possibility to further clarify the role played by Nrf2 as principal actor involved in cellular defence from oxidative stress, since Nrf2 not only tightly controls cellular redox status but it is also activated during ischemic stroke promoting the expression of antioxidant genes (Wang L. et al., Antioxidants 2022; Farina M et al., Molecules 2021; Liu L., Front. Pharmacol. 2019). Moreover, the activation of Nrf2 during stress conditions promote the activation of the transcriptional factor TFAM involved in the regulation of mitochondrial biogenesis. These findings further corroborate the key role of Nrf2 as sensor of oxidative stress and regulator of antioxidant defences in ischemic conditions. Therefore, we modified the contents of the lines 133-164 according to the relevance of Nrf2 in oxidative stress underlining its role in ischemic conditions. See Pag. 4, Lines 133-164 of the revised version of the manuscript
Comment # 3 “Are the contents on lines 231-252 associated with ischemic stroke? I think the contents just show the associations between oxidative stress and mitochondrial dysfunction independently of ischemic stroke.”
Answer: We agree with Referee that the contents on lines 231-252, are not directly correlated to ischemic stroke conditions. However, we believe that they are instrumental to support the dose and time dependence relationship among oxidative stress, mitochondrial dynamics and dysfunctions as it is widely described in ischemic conditions. See Pag. 6, Lines 243-255 and Pag.7, Lines 268-276
Comment # 4: “Ref. 7 is not appropriate because it is not accepted in a scientific journal. Ref. 7 is not suitable to cite in the review. Furthermore, ‘Data obtained in our laboratory demonstrated that mitochondrial dysfunction correlates with changes in mitochondrial morphology and mitochondrial quality control in hypoxic conditions [6,7]. (Lines 43-45)’ and relevant other sentences should be completely deleted.”
Answer: We are aware that the reference #7 refers to a manuscript submitted, however, the manuscript questioned by the Referee is under revision on Cell Death and Disease. Moreover, according to the guidelines of Antioxidants “Unpublished materials intended for publication” can be included in the manuscript and cited. See website: “Unpublished materials intended for publication:
- Author 1, A.B.; Author 2, C. Title of Unpublished Work (optional). Correspondence Affiliation, City, State, Country. year, status(manuscript in preparation; to be submitted).
- Author 1, A.B.; Author 2, C. Title of Unpublished Work. Abbreviated Journal Nameyear, phrase indicating stage of publication(submitted; accepted; in press).”
Comment # 5: “The authors cited Ref. 18 on the line 86. However, I could not find that neurodegenerative diseases are related to low antioxidant capacity in the affected brain to my best knowledge.”
Answer: We apologise for the mistake in citing the reference 18 in line 86, since, according to Referee observation it does not contain the information related to low antioxidant capacity of the brain that is intead reported in reference #17 cited in the manuscript (# 23 in the revised version of the manuscript): Chiurchiù V, Orlacchio A, Maccarrone M. Is Modulation of Oxidative Stress an Answer? The State of the Art of Redox Ther-apeutic Actions in Neurodegenerative Diseases. Oxid Med Cell Longev. 2016;2016:7909380: “Interestingly, along with increased ROS and RNS, direct examination of brain tissues from patients affected by neurodegenerative diseases also revealed a weakened cellular antioxidant defense, especially due to impairment and/or decrease of relevant antioxidants such as superoxide dismutase, catalase, glutathione/glutathione peroxidase, α-tocopherol, and uric acid”
See Pag.3, Line 89 Reference 23
Comment # 6: “The authors cited ref. 84 on line 284. However, ref. 84 does not indicate the main pathways involved in neuroprotective mechanisms include mitogen-activated protein kinases (MAPKs), the PI3K-Akt pathway and 283 the protein kinase C-ε (PKCε) one. Cite an appropriate reference.”
Answer: We thank the Referee for this important observation related to the inappropriate citation on lines 283 and 284. We apologise for the mistakes since the correct references are the number 85 and 86 (Now References 91, 92 in the revised version). In the new version of the manuscript the references are correctly cited. See Pag. 8 Line 302, References 91, 92
Comment # 7: Ref. 104 on the line 350 does not show ‘among them one assumes that NO interacts with prohibitin (PHB), an important protein localized in brain mitochondria that is involved in the regulation of mitochondrial morphology and dynamics.’
Answer: We thank the Referee for this important observation related to the inappropriate citation on line 350. We apologise for the mistake since the correct reference is the number 105 (Now Reference 106 in the revised version). In the new version of the manuscript the reference is correctly cited. See Pag.9 Line 378, Reference 106
Comment # 8. “The authors may regard both hypoxia and ischemia as the same pathological conditions. Ischemic condition is different from hypoxic one. Therefore, the authors should focus on brain ischemia when ischemic pathologies are discussed, while the authors should focus on brain (ischemia-)hypoxia when (ischemic and) hypoxic pathologies are discussed.”
Answer: We thank the Referee for this suggestion therefore, we focused on brain ischemia when discussed ischemic pathologies and on brain hypoxia when discussed hypoxic pathologies.
Comment # 9. “On lines 353-356, the authors mention that the E3-ubiquitin ligase Siah2 plays a key role in neuroprotection induced by ischemic preconditioning by orchestrating the balance between mitophagy and mitochondrial biogenesis in cortical neurons exposed to OGD/Reoxygenation. Does Siah2 plays a key role in neuroprotection via ubiquitination? Moreover, please appropriately cite after the above sentence. If Ref. 7 indicates the content of the above sentence, this is not the fact and the sentence have to be deleted.”
Answer: We thank the Referee for this important observation. In the revised version of the text we added further information about the role of Siah2 in hypoxic conditions and its ability to induce neuroprotection by ubiquitination of AKAP121 and NCX3, mitochondrial proteins (Sisalli et al., 2020). See Pag.9 Lines 381-389.
As far as concern the criticism related to the reference # 7, see previous answer to the comment # 4
Comment # 10. “In Table 1, low concentration of ROS or RNS may lead to neuroprotective effects, but high ROS or RNS concentration may lead to deleterious effects. However, TIA is short-term temporal brain ischemia with relatively mild decrease of cerebral blood flow. Therefore, not only low or high concentration but also the duration may affect the pathomechanisms. The authors should include the angle of the time duration.”
Answer: We agree with Referee’s observation that not only low or high concentration but also the duration of exposure to ROS and RNS plays a key role in neuroprotection/neurodegeneration as explained in the answer to comment #3, however, we think that to introduce the angle of the time duration in the table 2 might be confusing.”
Comment # 11. “Figure 2 appears to show that Siah2 and PHB may induce mitochondrial dynamics. However, contributions of Siah2 and PHB to mitochondrial fusion, fission, biogenesis, and mitophagy are not mentioned in the text.”
Answer: We thank the Referee for the observation related to the lack of information about the role of Siah2 and PHB in mitochondrial fusion, fission, biogenesis, and mitophagy. In the revised version of the manuscript we provide to clarify these concepts and added specific references. See Pag. 9, Lines 375-389.
Reviewer 2 Report
The manuscript by Dr. D’Apolito and colleagues titled "Oxidative metabolism in brain ischemia and preconditioning: the dual side of the coin" is a synthetic short review about detrimental and beneficial effects induced by oxidative metabolism induced by ischemic pre-conditioning.
The manuscript is a succint overview, which is well written and well organized.
I wish to congratulate the authors for their elegant efforts and recommend the manuscript for publication in its present form
No comments to provide
Author Response
Referee #2
No criticisms raised.
We thank the Referee for the positive evaluation
Reviewer 3 Report
Their main conclusion of this study ”Oxidative metabolism in brain ischemia and preconditioning: the dual side of the coin.” is to reveal that brain ischemia and ischemic preconditioning as two useful models to study the consequences of oxidative stress and mitochondrial dysfunction on neuronal survival. Moreover, the authors have discussed a great number of experiments in both ischemic models, harmful (ischemia) or protector (preconditioning or ischemic tolerance).
However, in order to strengthen their conclusions 1) they should discuss recent studies based on ischemic preconditioning models, both in vivo and in vitro. 2) Also, they should clearly explain specific points of the discussion. Specifically, in order to reinforce the role of ROS and RNS in the mitochondrial dysfunction and their relevance during short or prolonged time of ischemia.
Main Remarks:
The authors reviewed clearly the ideas but they did not provide sufficient recent references based on the topic:
RESULTS AND REFERENCES:
1. In the Introduction there are repeated ideas (lines 34-36), such as the concept of oxidative stress. They should be revised.
2. In page 2 the authors mentioned that: “oxidative stress may exert beneficial effects in those experimental conditions characterized by sublethal stress conditions”. They should provide evidence to support these beneficial effects.
3. The authors should add specific bibliography related to their revision. On line 57 of page 2 they have indicated “reproduced both in vivo and in vitro,” However, they have not indicated any reference.
4. The authors should review the main recent factors involved in the ischemic cascade or induced by ischemic preconditioning. I suggest that authors could elaborate a table with this information. They should discuss if these factors are equal in both models?
CONCLUSIONS:
In conclusions, on lines 376-378, the authors mentioned that “short and repeated periods of O2 deprivation, as those occurring during ischemic preconditioning, are able to improve mitochondrial dynamics stimulated by low levels of ROS and RNS production, and to promote neuroprotection” the authors should discuss which is the IPC-mediated mechanism inducing mitochondrial benefit?
DISCUSSION:
1. In table 1, in page 4, the authors reviewed the dual role of ROS and RNS. What is the range of low or higher? It depends on the reactive species nature? i.e. They should discuss if higher concentrations favour necrosis while lower levels promote apoptosis.
2. On page 5, What levels of calcium are induced after IPC or ischemia? Is it very different under both stimuli? What is the role of intracellular calcium in IPC-mediated protection?
3. On page 6, The mitochondria dynamic is different under IPC compared to prolonged ischemia injury? The explained content in Fig 1 occurs in both conditions? And is it true how the IPC is capable of exerting cytoprotection? What is the possible mechanism?
4. In page 8: The authors indicated that E3-ubiquitin ligase Siah2 plays a key role in neuroprotection induced by ischemic preconditioning. Do the authors know other ubiquitin-proteasome systems involved in IPC-induced neuroprotection? The authors should review results obtained in different systems of protein control stability.
1. In the abstract the authors should write superoxide radical in a correct form O2•−
2. The authors should review words in bold, i.e. a great (page 1); a process (page 3; thus…
3. In line 94, they should indicate the correct form of hydroperoxyl radical (HOO-).
Author Response
REFEREE N 3
Comment #1: “In the abstract the authors should write superoxide radical in a correct form O2•−“
Answer: We apologize for the mistake in indicating superoxide radical that has been now corrected in the revised version of the manuscript. See Pag 1, Line: 22
Comment #2: “In the Introduction there are repeated ideas (lines 34-36), such as the concept of oxidative stress should be revised”
Answer: We thank the Referee for this suggestion. In the revised version of the manuscript we removed the sentence repeating the concept of oxidative stress. See page 1; lines 35-36.
Comment # 3: “In page 2 the authors mentioned that: “oxidative stress may exert beneficial effects in those experimental conditions characterized by sublethal stress conditions”. They should provide evidence to support these beneficial effects”.
Answer: We thank the Referee for this suggestion. In the revised version of the manuscript we provided evidence to support the beneficial effect of oxidative stress in experimental conditions reproducing sublethal stress. See Pag. 2, Lines 61-62, References # 13-16.
Comment # 4 “The authors should add specific bibliography related to their revision. On line 57 of page 2 they have indicated “reproduced both in vivo and in vitro,” However, they have not indicated any reference”
Answer: We apologise for the missing references on line 57 page 2. In the revised version of the manuscript we provided the references required. See Page: 2; Lines: 66-67, References 18-21.
Comment #5: “The authors should review the main recent factors involved in the ischemic cascade or induced by ischemic preconditioning. I suggest that authors could elaborate a table with this information. They should discuss if these factors are equal in both models?”
Answer: We thank the Referee for this important observation. According to Referee’s suggestion in the revised version of the manuscript we included a table summarising the main factors involved in the ischemic cascade or induced by ischemic preconditioning, and specified those factors that play a detrimental or neuroprotective effects in the above mentioned conditions. We also included specific references and discussed the different role of these factors in section 3 and 4. See Table 1 of the new version of the manuscript. Page 2, Line 72; Pages 5 (section 3) and 7 (section 4).
Comment #6: “In conclusions, on lines 376-378, the authors mentioned that “short and repeated periods of O2 deprivation, as those occurring during ischemic preconditioning, are able to improve mitochondrial dynamics stimulated by low levels of ROS and RNS production, and to promote neuroprotection” the authors should discuss which is the IPC-mediated mechanism inducing mitochondrial benefit?”
Answer: We thank the Referee for this suggestion. In the revised version of the manuscript we indicated figure 2 to describe IPC-mediated mechanisms involved in mitochondrial benefit and discussed the mechanisms inducing mitochondrial beneficial effects in paragraph 4. See Pag. 10; Lines 409-414.
Comment #7: “In table 1, in page 4, the authors reviewed the dual role of ROS and RNS. What is the range of low or higher? It depends on the reactive species nature? They should discuss if higher concentrations favour necrosis while lower levels promote apoptosis”.
Answer: We thank the Referee for this important observation that has been addressed in the revised version of the manuscript. Indeed, the range of ROS and RNS is directly correlated to cellular and to antioxidant mechanism integrity. Therefore, depending on the severity of disruption of the antioxidant defence, more pronounced in ischemia than in ischemic preconditioning, some intracellular pathways sensitive to ROS production can be activated to promote detrimental (necrosis/apoptotic cell death)/beneficial (cell survival) effects respectively. See page 4, Lines 165-169.
Comment #8: “On page 5, What levels of calcium are induced after IPC or ischemia? Is it very different under both stimuli? What is the role of intracellular calcium in IPC-mediated protection?”
Answer: We thank the Referee for the suggestion to clarify the aspect related to calcium concentration in ischemia, and in IPC. In the revised version of the manuscript we underlined these differences mainly consisting in: 1. abnormal calcium overload in ischemic conditions responsible for detrimental effects on cellular survival, and 2. in low calcium increases in IPC responsible for activation of intracellular events responsible for neuroprotection. See Pag.5. Lines: 200-214; Pag 6 Lines 215-222; Pag. 9, Lines 349-372.
Comment #9: ”On page 6, The mitochondria dynamic is different under IPC compared to prolonged ischemia injury? The explained content in Fig 1 occurs in both conditions? And is it true how the IPC is capable of exerting cytoprotection? What is the possible mechanism?”
Answer: We thank the Referee for the suggestion to clarify the different role played by mitochondrial quality control in neuroprotection induced by IPC versus ischemic injury. In this regard, in the revised version of the manuscript we better clarified that the mitochondrial dynamics are different under IPC compared to ischemic injury. As reported in Fig1 mitochondrial fragmentation plays a key role in the detrimental effects occurring in ischemic conditions since, fragmented mitochondria are dysfunctional and thus committed to mitophagy without generation of new functional mitochondria. Conversely, during IPC conditions, mitochondrial fragmentation causes mitophagy and subsequently activates biogenesis through the induction of E3 ubiquitin ligase Siah2, which is sensitive to O2 deprivation. Therefore, the activation of biogenesis in IPC conditions exerts neuroprotection. See Pag. 9, Lines 381-387; Pag. 10; Lines 402-411.
Comment #10: “In page 8: The authors indicated that E3-ubiquitin ligase Siah2 plays a key role in neuroprotection induced by ischemic preconditioning. Do the authors know other ubiquitin-proteasome systems involved in IPC-induced neuroprotection? The authors should review results obtained in different systems of protein control stability”.
Answer: We thank the Referee for this interesting observation. Although many other E3 ubiquitin ligases might be activated by ischemic insult, and probably play a role in IPC, however their description is out of scope of the present review.
Comment #9: ”On page 6, The mitochondria dynamic is different under IPC compared to prolonged ischemia injury? The explained content in Fig 1 occurs in both conditions? And is it true how the IPC is capable of exerting cytoprotection? What is the possible mechanism?”
Answer: We thank the Referee for the suggestion to clarify the different role played by mitochondrial quality control in neuroprotection induced by IPC versus ischemic injury. In this regard, in the revised version of the manuscript we better clarified that the mitochondrial dynamics are different under IPC compared to ischemic injury. As reported in Fig1 mitochondrial fragmentation plays a key role in the detrimental effects occurring in ischemic conditions since, fragmented mitochondria are dysfunctional and thus committed to mitophagy without generation of new functional mitochondria. Conversely, during IPC conditions, mitochondrial fragmentation causes mitophagy and subsequently activates biogenesis through the induction of E3 ubiquitin ligase Siah2, which is sensitive to O2 deprivation. Therefore, the activation of biogenesis in IPC conditions exerts neuroprotection. See Pag. 9, Lines 381-387; Pag. 10; Lines 402-411.
Comment #10: “In page 8: The authors indicated that E3-ubiquitin ligase Siah2 plays a key role in neuroprotection induced by ischemic preconditioning. Do the authors know other ubiquitin-proteasome systems involved in IPC-induced neuroprotection? The authors should review results obtained in different systems of protein control stability”.
Answer: We thank the Referee for this interesting observation. Although many other E3 ubiquitin ligases might be activated by ischemic insult, and probably play a role in IPC, however their description is out of scope of the present review.
Reviewer 4 Report
Dear Authors,
I have read with a great interest and pleasure your review "Oxidative metabolism in brain ischemia and preconditioning: the dual side of the coin." You have elegantly explained the mechanisms and the role of oxidative stress in brain injury (mostly, in ischemic brain injury). Moreover, you have proven that oxidative stress and presence of ROS and RNS is not "pure evil" as many scientists (and clinicians) think. I think this issues make your manuscript very valuable.
Plese, find below my suggestions that may improve your paper with aging some additional attention from the clinicians:
1. I do miss a paragraph focused on anti-oxidative therapeutical mechanisms, especially with usage of pharmacological antioxidants, e.g. melatonin. I thing that short description of the role of such antioxidants would be interesting.
2. I think that in context of preconditioning ischemia, some attention should be given to the suggested neuroprotective role (both in neurodegenerative and vascular brain injury) of tobacco smoking. I think that may be considered as a form of ischemic/hypoxic pre-conditioning. I would suggest adding some comments on that.
Dear Authors,
As my suggestions are of general nature, I do not have precise comments to be put in here.
Author Response
REFEREE # 4
Comment #1. “I do miss a paragraph focused on anti-oxidative therapeutically mechanisms, especially with usage of pharmacological antioxidants, e.g. melatonin. I think that short description of the role of such antioxidants would be interesting”.
Answer: We thank the Referee for the positive evaluation of the present manuscript and also for the appropriate observation about the importance to discuss the role of antioxidant therapeutically strategies related to the subject of review, including melatonin. In the revised version of the manuscript a short mention to the topic and a Table including the antioxidants have been added. See Page 11, Lines 429-441, and Table 3.
Comment #2. “I think that in context of preconditioning ischemia, some attention should be given to the suggested neuroprotective role (both in neurodegenerative and vascular brain injury) of tobacco smoking. I think that may be considered as a form of ischemic/hypoxic pre-conditioning. I would suggest adding some comments on that”
Answer: We thank the Referee for this suggestion that has been introduced in the revised version of the manuscript. See Pag: 11, Lines: 435-442
Round 2
Reviewer 1 Report
In the authors' reply, the locations of revised text are not precisely indicated. Most of the locations may be wrong. Please indicate precise numbers of lines or pages so that I can effectively review the revised manuscript.
None.
Author Response
REFEREE #1
Comment # 1: “On the line 99-101, ‘Moreover, different environmental conditions, such as radiation or pollution contribute to the increase in ROS production responsible for cellular detrimental effects [2, 24].’ However, it is unclear the connection between the above sentence and anteroposterior sentences. Consider delete them.”
Answer: We thank the Referee for the observation. In the revised version we indicated in green the modified line. See page 3, line 102 of the revised version of the manuscript
Comment # 2: The contents on lines 133-164 are not applied to the following sections. I do not believe the contents are necessary to be described in the review. Consider delete them.
Answer: We thank the Reviewer for the valuable observation on the contents described in Lines 133-164 of the manuscript that gave us the possibility to further clarify the role played by Nrf2 as principal actor involved in cellular defence from oxidative stress, since Nrf2 not only tightly controls cellular redox status but it is also activated during ischemic stroke promoting the expression of antioxidant genes (Wang L. et al., Antioxidants 2022; Farina M et al., Molecules 2021; Liu L., Front. Pharmacol. 2019). Moreover, the activation of Nrf2 during stress conditions promote the activation of the transcriptional factor TFAM involved in the regulation of mitochondrial biogenesis. These findings further corroborate the key role of Nrf2 as sensor of oxidative stress and regulator of antioxidant defences in ischemic conditions. Therefore, we modified the contents of the lines 133-164 according to the relevance of Nrf2 in oxidative stress underlining its role in ischemic conditions. See Pag. 4, Lines 133-164 of the revised version of the manuscript highlighted in green.
Comment # 3 “Are the contents on lines 231-252 associated with ischemic stroke? I think the contents just show the associations between oxidative stress and mitochondrial dysfunction independently of ischemic stroke.”
Answer: We agree with Referee that the contents on lines 231-252, are not directly correlated to ischemic stroke conditions. However, we believe that they are instrumental to support the dose and time dependence relationship among oxidative stress, mitochondrial dynamics and dysfunctions as it is widely described in ischemic conditions. See Pag. 6, Lines 243-255 and Pag.7, Lines 268-276 now highlighted in green.
Comment # 4: “Ref. 7 is not appropriate because it is not accepted in a scientific journal. Ref. 7 is not suitable to cite in the review. Furthermore, ‘Data obtained in our laboratory demonstrated that mitochondrial dysfunction correlates with changes in mitochondrial morphology and mitochondrial quality control in hypoxic conditions [6,7]. (Lines 43-45)’ and relevant other sentences should be completely deleted.”
Answer: We are aware that the reference #7 refers to a manuscript submitted, however, the manuscript questioned by the Referee is under the first step revision on Cell Death and Disease. Moreover, according to the guidelines of Antioxidants “Unpublished materials intended for publication” can be included in the manuscript and cited. See website: “Unpublished materials intended for publication:
- Author 1, A.B.; Author 2, C. Title of Unpublished Work (optional). Correspondence Affiliation, City, State, Country. year, status(manuscript in preparation; to be submitted).
- Author 1, A.B.; Author 2, C. Title of Unpublished Work. Abbreviated Journal Nameyear, phrase indicating stage of publication(submitted; accepted; in press).”
Comment # 5: “The authors cited Ref. 18 on the line 86. However, I could not find that neurodegenerative diseases are related to low antioxidant capacity in the affected brain to my best knowledge.”
Answer: We apologise for the mistake in citing the reference 18 in line 86, since, according to Referee observation it does not contain the information related to low antioxidant capacity of the brain that is intead reported in reference #17 cited in the manuscript (# 23 in the revised version of the manuscript): Chiurchiù V, Orlacchio A, Maccarrone M. Is Modulation of Oxidative Stress an Answer? The State of the Art of Redox Ther-apeutic Actions in Neurodegenerative Diseases. Oxid Med Cell Longev. 2016;2016:7909380: “Interestingly, along with increased ROS and RNS, direct examination of brain tissues from patients affected by neurodegenerative diseases also revealed a weakened cellular antioxidant defense, especially due to impairment and/or decrease of relevant antioxidants such as superoxide dismutase, catalase, glutathione/glutathione peroxidase, α-tocopherol, and uric acid”
See Pag.3, Lines 88-89 Reference 23 of the revised version of the manuscript now highlighted in green.
Comment # 6: “The authors cited ref. 84 on line 284. However, ref. 84 does not indicate the main pathways involved in neuroprotective mechanisms include mitogen-activated protein kinases (MAPKs), the PI3K-Akt pathway and 283 the protein kinase C-ε (PKCε) one. Cite an appropriate reference.”
Answer: We thank the Referee for this important observation related to the inappropriate citation on lines 283 and 284. We apologise for the mistakes since the correct references are the number 85 and 86 (Now References 91, 92 in the revised version). In the new version of the manuscript the references are correctly cited. See Pag. 8 Line 302, References 91, 92 highlighted in green.
Comment # 7: Ref. 104 on the line 350 does not show ‘among them one assumes that NO interacts with prohibitin (PHB), an important protein localized in brain mitochondria that is involved in the regulation of mitochondrial morphology and dynamics.’
Answer: We thank the Referee for this important observation related to the inappropriate citation on line 350. We apologise for the mistake since the correct reference is the number 105 (Now Reference 106 in the revised version). In the new version of the manuscript the reference is correctly cited. See Pag.9 Lines 377-378, Reference 106, highlighted in green.
Comment # 8. “The authors may regard both hypoxia and ischemia as the same pathological conditions. Ischemic condition is different from hypoxic one. Therefore, the authors should focus on brain ischemia when ischemic pathologies are discussed, while the authors should focus on brain (ischemia-)hypoxia when (ischemic and) hypoxic pathologies are discussed.”
Answer: We thank the Referee for this suggestion therefore, we focused on brain ischemia when discussed ischemic pathologies and on brain hypoxia when discussed hypoxic pathologies.
Comment # 9. “On lines 353-356, the authors mention that the E3-ubiquitin ligase Siah2 plays a key role in neuroprotection induced by ischemic preconditioning by orchestrating the balance between mitophagy and mitochondrial biogenesis in cortical neurons exposed to OGD/Reoxygenation. Does Siah2 plays a key role in neuroprotection via ubiquitination? Moreover, please appropriately cite after the above sentence. If Ref. 7 indicates the content of the above sentence, this is not the fact and the sentence have to be deleted.”
Answer: We thank the Referee for this important observation. In the revised version of the text we added further information about the role of Siah2 in hypoxic conditions and its ability to induce neuroprotection by ubiquitination of AKAP121 and NCX3, mitochondrial proteins (Sisalli et al., 2020). See Pag.9 Lines 381-389, highlighted in green.
As far as concern the criticism related to the reference # 7, see previous answer to the comment # 4
Comment # 10. “In Table 1, low concentration of ROS or RNS may lead to neuroprotective effects, but high ROS or RNS concentration may lead to deleterious effects. However, TIA is short-term temporal brain ischemia with relatively mild decrease of cerebral blood flow. Therefore, not only low or high concentration but also the duration may affect the pathomechanisms. The authors should include the angle of the time duration.”
Answer: We agree with Referee’s observation that not only low or high concentration but also the duration of exposure to ROS and RNS plays a key role in neuroprotection/neurodegeneration as explained in the answer to comment #3, however, we think that to introduce the angle of the time duration in the table 2 might be confusing.”
Comment # 11. “Figure 2 appears to show that Siah2 and PHB may induce mitochondrial dynamics. However, contributions of Siah2 and PHB to mitochondrial fusion, fission, biogenesis, and mitophagy are not mentioned in the text.”
Answer: We thank the Referee for the observation related to the lack of information about the role of Siah2 and PHB in mitochondrial fusion, fission, biogenesis, and mitophagy. In the revised version of the manuscript we provide to clarify these concepts and added specific references. See Pag. 9, Lines 375-389, highlighted in green.
Reviewer 3 Report
Thank you for accepting my suggestions.
NA
Author Response
We thank the Referee for the positive comment to the revisioned manuscript
Round 3
Reviewer 1 Report
The manuscript have been improved.
The manuscript have been improved.